Altitudinal gradients in Magellanic sub-Antarctic lagoons: the effect of elevation on freshwater macroinvertebrate diversity and distribution

Rendoll Cárcamo Javier javier.rendoll@gmail.com 1 2 3
Contador Tamara 1 2 3 4
Gañán Melisa 1
Pérez Troncoso Carolina 1 2 3 4
Maldonado Márquez Alan 1 2 3 4
Convey Peter 5
Kennedy James 1 2 6
Rozzi Ricardo 2 3 7
1 Wankara Sub-Antarctic and Antarctic Freshwater Ecosystems Laboratory, Universidad de Magallanes , Puerto Williams , Magallanes , Chile
2 Subantarctic Biocultural Conservation Program, Omora Ethnobotanical Park, Universidad de Magallanes , Puerto Williams , Magallanes , Chile
3 Institute of Ecology and Biodiversity, Universidad de Chile , Santiago , Chile
4 Núcleo Milenio de Salmónidos Invasores, INVASAL, Iniciativa Científica Milenio, ICM, Núcleo Científico Milenio , Concepción , Chile
5 British Antarctic Survey, NERC , Cambridge , United Kingdom
6 Department of Biological Sciences, University of North Texas , Denton , United States of America
7 Department of Philosophy and Religion, University of North Texas , Denton , United States of America
Costello Mark
Electronic publication date: 2019 Jul 29
Publication date: 2019
Volume: 7
Electronic Location ID: e7128
Received 2019 Mar 1; Accepted 2019 May 15
Copyright: ©2019 Rendoll Cárcamo et al.
Copyright year: 2019
Copyright holder: Rendoll Cárcamo et al.
License: This is an open access article distributed under the terms of the Creative Commons Attribution License, which permits unrestricted use, distribution, reproduction and adaptation in any medium and for any purpose provided that it is properly attributed. For attribution, the original author(s), title, publication source (PeerJ) and either DOI or URL of the article must be cited.
License URL: https://creativecommons.org/licenses/by/4.0/

Keywords: Elevation gradient, Sub-antarctic, Littoral invertebrates, Lakes, Southern Chile

Funding: Institute of Ecology and Biodiversity (IEB) ICM, P05-002 CONICYT PFB-23 FONDECYT Project 11130451 NERC core funding to the BAS ‘Biodiversity, Evolution and Adaptations’ Team Iniciativa Científica Milenio Invasive Salmonids INVASAL INACH Grant RT 48-16 Javier Rendoll Cárcamo was supported by a master’s scholarship granted by the Institute of Ecology and Biodiversity (IEB), by ICM, P05-002 and CONICYT PFB-23 projects. Research was funded by FONDECYT Project 11130451, led by Tamara Contador. Peter Convey is supported by NERC core funding to the BAS ‘Biodiversity, Evolution and Adaptations’ Team. Tamara Contador is supported by INACH grant RT 48-16, and by Iniciativa Científica Milenio Invasive Salmonids INVASAL. The funders had no role in study design, data collection and analysis, decision to publish, or preparation of the manuscript.

==============================
Background

The study of altitudinal gradients provides insights about species diversity, distribution patterns and related drivers. The Magellanic sub-Antarctic ecoregion has a steep elevational gradient, peaking at around 1,000 m a.s.l., and marked changes in temperature and landscape composition can be observed over relatively short distances.

Methods

This study assessed freshwater macroinvertebrate diversity associated with lakes and ponds along the altitudinal gradient of a Magellanic sub-Antarctic watershed.

Results

A monotonic decline in species richness was observed with increasing elevation, with simpler and more even community composition at higher altitude. This pattern differs from the mid-peak trend found in streams of the same watershed. Functional feeding group structure also diminished with increasing elevation.

Discussion

The study provides a descriptive baseline of macroinvertebrate community structure associated with lentic freshwater ecosystems in the Magellanic sub-Antarctic ecoregion, and confirms that elevation has substantial effects on community structure, function and environmental features, even in these relatively low elevation mountain ranges. The harsh environmental conditions of this ecoregion increase freshwater macroinvertebrate development time, as well as decreasing habitat availability and food supply, supporting simple but well adapted communities. In conjunction with previous research, this study provides a watershed-scale platform of information underpinning future long-term research in the region.

Introduction

Since the times of the early naturalists such as Linnaeus, Humboldt, Darwin and Wallace, understanding how living organisms distribute and change along spatial gradients has been key for the development of theories about their origins and diversification (Lomolino, 2001). From the wide spectrum of related variables, temperature is one of the most studied drivers of biodiversity richness and distribution (De Mendoza & Catalán, 2010; McCain & Grytnes, 2010). Air temperature predictably decreases as elevation increases, and at the same time regulates water temperature, ice and snow cover (Čiamporová Zat’ovičová et al., 2010). In particular, latitudinal and elevational gradients can be considered analogues in terms of associated environmental clines and ecological features (Bozinovic, Calosi & Spicer, 2011; Rahbek, 1995). In this context, faunal (and floral) research along elevational gradients is of critical importance in the study of global biodiversity drivers (McCain & Grytnes, 2010).

Increasing anthropogenic impact on the environment can significantly alter thermal regimes of freshwater ecosystems (Bates, Kundzewicz & Wu, 2008), threatening their biota (Millennium Ecosystem Assessment, 2005). It is estimated that at least 126,000 invertebrate species inhabit freshwater ecosystems (Balian et al., 2008), of which between 10 and 15% are threatened by extinction or are already extinct (Vörösmarty et al., 2010). Understanding invertebrate sensitivity to ecological changes is of great importance, and these organisms have proven to be useful bioindicators (Kremen et al., 1993). Environmental temperature influences distribution, fitness, physiology and, ultimately, the survival of invertebrates (Boggs, 2016), and research into its influence is a priority to underpin conservation and management strategies (Contador, Kennedy & Rozzi, 2012).

Southern South American ecosystems were strongly influenced by ice sheet expansion leading up to the Last Glacial Maximum (23,000–19,000 years before present), resulting in a heterogeneous and diverse mosaic of contemporary habitats (Hulton et al., 2002; Glasser et al., 2008). Glacial erosion shaped numerous archipelagos, mountain ranges and valleys, across numerous watersheds. In addition, the land:ocean ratio in the 40–60°S latitude belt is about 2:98, compared to 54:46 at equivalent latitudes in the Northern Hemisphere, supporting oceanic buffered climatic conditions (Rozzi et al., 2012). The Magellanic sub-Antarctic ecoregion lies between 47–56°S, and is an area characterized by low altitude mountainous terrain (∼1,000 m a.s.l.), with short, steep and marked environmental gradients (Contador et al., 2015). In addition, the buffering effect of the ocean on temperature reduces drastically with elevation, which decreases by approximately 1 °C for each 100 m increase in elevation (Contador et al., 2015; Méndez, Rozzi & Cavieres, 2013). While this ecoregion is considered one of the last pristine areas left in the world (Mittermeier et al., 2003), it is not free from the effects of global environmental change, massive tourism and introduced species (Rozzi et al., 2012).

The freshwater macroinvertebrate species found within this ecoregion are highly endemic and well adapted to the cold-temperate weather conditions (e.g., through demonstrating variable voltinism patterns and physiological optima) (Contador et al., 2014; Contador & Kennedy, 2016). However, although much effort has been devoted to studying ecological patterns in running waters across steep altitudinal gradients (see Contador et al., 2015; Moorman et al., 2006), lentic ecosystems in this region remain understudied.

The primary goals of this study, therefore, were to (1) provide a concise description of the richness and composition of freshwater macroinvertebrate communities occurring in lakes and ponds across the altitudinal gradient of a Magellanic sub-Antarctic watershed, and (2) to compare macroinvertebrate assemblage composition and functional feeding groups throughout the gradient. The results from this study complement previous descriptive studies of stream fauna, together with which they contribute to a baseline for future long-term research and monitoring of southern South American freshwater ecosystems.

Materials & Methods

Study area

The Magellanic sub-Antarctic ecoregion, located in southern Chile, is characterized by watersheds with acute environmental clines, diverse vegetation profiles, and the presence of a wide variety of habitats and microhabitats over a short elevational range (Pisano, 1977; Contador et al., 2015). This ecoregion is part of the South American forest biome, harboring the largest forest and wetland areas of the Southern Hemisphere. Embedded within the ecoregion is the Cape Horn Biosphere Reserve (CHBR; Fig. 1), an area designated to protect the ecoregion from the pressures of global change (Rozzi et al., 2006). Furthermore, this region contains some of the world’s cleanest rainwater, as it is located to the south of the typical tracks of industrial-polluted wind currents (Hedin, Armesto & Johnson, 1995; Weathers et al., 2000; Mach et al., 2016). Navarino Island lies in the southern part of this ecoregion. The mountain ranges in Navarino Island are characterized by short and steep altitudinal gradients (rising to ∼1,000 m a.s.l.), with associated changes in air and water temperatures with elevation (Contador et al., 2015). Annual mean air temperature is 5.7 °C close to sea level, decreasing to 0 °C at 728 m a.s.l. (Méndez, Rozzi & Cavieres, 2013). This study was conducted in lagoons (lakes and ponds) located along the altitudinal gradient of the Róbalo River watershed, which provides drinking water to Puerto Williams, the world’s southernmost town. The annual average water temperature in the running waters of the watershed is 5.7 °C at 120 m and only 1 °C at 586 m a.s.l. (Contador et al., 2015).

Figure 1 Cape Horn Biosphere Reserve and study sites on Navarino Island.

Location of the Cape Horn Biosphere Reserve in Southern South America. Navarino Island and study sites from the Róbalo river watershed are zoomed. CL, Castor lagoon; RL, Róbalo lake; SL, el Salto lagoon; BL, Bandera lagoon.

Study site selection, habitat characterization and physico-chemical parameters

This study was conducted in lagoons located throughout the altitudinal gradient of the Róbalo river watershed (Table 1, Fig. 1). These lagoons are surrounded by evergreen forests and Sphagnum peat bogs, mixed forests and shrub patches, deciduous forests near the tree line (Krummholz), and high Andean tundra composed of cushion plants, mosses and lichens. Four lagoons were selected, located at different altitudinal levels, thus allowing for comparison between the different zones found within the gradient. Along with the physical description of each lagoon, the surrounding habitats were characterized. Within each lagoon, four different habitat types were identified based on substrate composition, and the presence/absence and type of vegetation (vascular or non-vascular): (i) rock bottoms (gravels from 1 to 10 cm), (ii) submerged vegetation (aquatic or submerged vascular plants), (iii) debris (leaf litter, roots, or logs), and (iv) aquatic mosses (completely or partially submerged). Temperature, pH and conductivity were measured three times in each habitat type on each sampling occasion (n = 3), using a multiparameter sensor (Conductivity pH TDS Hanna Tester HI98130). Additionally, HOBO® data loggers (model U22 Water Temperature Pro version 2) were permanently installed in each lagoon (anchored at 20 cm depth at a randomly selected location) from February 2015 until March 2016, recording water temperature every 4 h. Mean, maximum and minimum daily and monthly temperatures (°C) were calculated using the data obtained.

Table 1 Study sites.

Physical description of the four sampled lagoons along the elevation gradient of the Róbalo river watersed, Navarino Island, Chile.

Site	Name	Georeference	Elevation (m.a.s.l)	Water inflow and/or outflow	Adjacent vegetation	
I	Castor lagoon	54°56′19″S/67°38′15″W	20	Diffuse	Evergreen forests and Sphagnum peatlands	
II	Róbalo lake	54°58′12″S/67°40′29″W	250	South located inflow, outflow at North-East	Mixed deciduous and evergreen forests and scrublands	
III	El Salto lagoon	54°59′26″S/67°40′56″W	480	South located inflow, outflow at North	Scrub forests, mosses and herbs	
IV	Bandera lagoon	54°58′26″S/67°38′41″W	700	South-East located inflow, outflow at West	Mosses, lichens and cushion plants	

Biological sampling and macroinvertebrate identification

During the 2015 austral summer (January, February) freshwater macroinvertebrates were collected. From each lagoon three samples were taken in each habitat with a D-frame net (150 µm mesh). Each sample consisted of 1 min of net sweeping (standard sampling), and a complementary collection to obtain a comparable and representative sample of each habitat (Table 2). The latter samples were analyzed with each standard sample. Macroinvertebrates were stored in flasks in 70% ethanol, and returned to the laboratory for sorting, identification to the lowest taxonomical level possible and functional feeding group assignment according to the available literature (Domínguez & Fernández, 2009; Flint, Holzenthal & Harris, 1999; Libonatti, Michat & Torres, 2011; McLellan & Swick, 2007; Moorman et al., 2006; Siddall & Borda, 2004; Von Ellenrieder, 2003). Field and sampling permits were provided by Omora Ethnobotanical Park, and field collection of specimens were approved by Universidad de Magallanes (certificate number: no 80/CEC/2018).

Table 2 Sampling methods.

Standard and complementary sampling for each microhabitat in each lagoon at Róbalo river watershed, Navarino Island, Chile. Sampling time unit corresponds to one minute.

Habitat type	Standard sampling	Complementary sampling	
Rock bottoms	D-frame net per time unit	Rinse and dislodge organisms from rocks	
Submerged vegetation	D-frame net per time unit	25 cm2 to 64 cm2 plant collection (according microhabitat availability)	
Debris	D-frame net per time unit	200 ml flask full of litter, or rinse and dislodge organisms from roots, branches and logs	
Aquatic mosses	D-frame net per time unit	25 cm2 to 64 cm2 moss collection (according microhabitat availability)	

Data analyses

Abiotic data were tested for normality and found to be non-normally distributed (Shapiro–Wilks test). The non-parametric Kruskal–Wallis (K–W) one way analysis of variance was therefore used. When significance was achieved, Wilcoxon post hoc pairwise comparisons were made. Diversity metrics were calculated for each lagoon along the altitudinal gradient. We used absolute and mean richness to describe taxonomic richness at each elevation, Shannon–Wiener diversity index (H′), Peilou’s evenness (J′) and N1 of Hill’s number series (Shannon–Wiener equivalent for comparison) to assess community structure (Hill, 1973; Magurran, 1988). Community structure indices were calculated using Primer 5.0 (Clarke, 1993). Changes in macroinvertebrate assemblage richness, abundance, diversity metrics and composition were tested with the non-parametric permutation test PERMANOVA (Anderson, 2001). Prior to PERMANOVA analyses, the PERMDISP test was performed to assess homogeneity within and between groups (Anderson, 2005). Euclidean distance between pairs of observations was calculated for the univariate analyses (Claudet et al., 2006). For multivariate analyses, Bray–Curtis dissimilarity matrices were calculated between pairs of observations and data were log(x + 1) transformed. All PERMANOVA analyses were performed using FORTRAN software (Anderson, 2005). For univariate and multivariate analyses we used a factorial design, considering elevation and habitat type. To determine specific importance of macroinvertebrates at each site, and their associated functional feeding group, we used the Similarity Percentages multivariate analysis SIMPER (Primer 5.0, Clarke, 1993), with a 90% contribution cutoff.

Results

Altitudinal abiotic parameters

The sampled lagoons remained surface frozen for between four and seven months, with water temperature dropping to around 0 °C in the first weeks of May (austral autumn) and subsequently, depending on elevation, beginning to thaw in mid-September (austral spring) to mid-December (austral summer). The maximum average temperature was recorded in Castor lagoon (20 m a.s.l.), reaching 16.99 °C in December, while the minimum average temperature recorded was −0.19 °C in June in Róbalo lake (Fig. 2).

Figure 2 Water temperature profiles.

Lagoon monthly water temperature profiles retrieved from data loggers. Different color lines indicate a different lagoon.

Conductivity showed significant differences along the altitudinal gradient (K–W, P < 0.0001, Table S1) associated with the increase of elevation, ranging from 147.3 to 49.5 µS (Table 3). Significant differences in pH were detected between Castor lagoon and the remaining three sampled lagoons along the gradient (K–W, P < 0.0001). This low elevation lagoon showed acidic values, while the others were close to neutral pH. Water temperature recorded in the sampling events showed significant variation between elevations (K–W, P < 0.0001, Table S1), decreasing with increasing altitude.

Table 3 Abiotic parameters.

Mean values and non parametric Kruskal–Wallis analysis of variance and for abiotic parameters measured at sampling sites in the Róbalo river watershed, Navarino Island, Chile.

	Elevation (m.a.s.l.) and mean values (± SE)	H	p	
Parameter	20	250	480	700			
Conductivity (µS/cm)	147.3 (1.23)	67.25 (3.89)	57.08 (2.31)	49.5 (6.84)	41.267	<0.0001*	
pH	5.97 (0.09)	7.67 (0.1)	7.83 (0.22)	7.61 (0.21)	30.8702	<0.0001*	
Temperature (°C)	11.03 (0.11)	9.71 (0.09)	6.46 (0.19)	5.37 (0.17)	44.4119	<0.0001*	
Notes.

* Values with asterisks represent significant differences (p < 0.05).

Altitudinal diversity metrics

Taxonomic richness

A total of 9384 macroinvertebrates were collected in the study, from which we identified 28 taxa belonging to 13 orders and 17 families. Of the 28 taxa, 16 were identified to genus/species level, and the remaining to order, family or subfamily level, catalogued as morpho-species (Table S2). Sites at lower elevations showed the highest absolute richness, S = 18 at 20 m and S = 17 at 250 m a.s.l., while in sites at the higher elevations we recorded fewer taxa, S = 11 at 480 m and S = 6 at 700 m a.s.l.

Altitudinal variation in richness, abundance and diversity metrics

The permutation analysis detected significant differences in mean macroinvertebrate richness between elevations and habitats (PERMANOVA, Elev F = 39.695, P = 0.0002; Hab F = 5.2073, P = 0.0056, Fig. 3, Table S3). Castor lagoon, at 20 m a.s.l., had a significantly higher mean richness than the other elevations. Pairwise comparisons detected differences between all elevations, with the exception of 250 and 480 m a.s.l. (pairwise comparison, P = 0.5779, Table S4). Significant differences were also detected in macroinvertebrate mean abundance along the elevation gradient (Fig. 3), between the habitats and in interactions between factors (PERMANOVA, Elev F = 56.544, P = 0.0002; Hab F = 12.54, P = 0.0002; Elev × Hab F = 10.515, P = 0.0002, Table S3). Mean abundance was significantly different between all elevations, between rock bottom habitats and the other three habitat types (pairwise comparisons in Table S4). Differences in diversity (H′, N1) and evenness (J′) indexes were also found with elevation, showing a similar pattern with values decreasing as elevation increased (PERMANOVA, Shannon index, F = 31.566, P = 0.0002; Hill’s N1 F = 25.478, P = 0.0002, Pielou’s evenness F = 18.156, P = 0.0002, Table S3), and indicating a more uniform community at 700 m a.s.l. in Bandera lagoon. However, significant differences in mean abundance between habitats were found for the Shannon diversity index and Pielou’s evenness (PERMANOVA, Shannon index F = 3.4004, P = 0.0294; Pielou’s evenness F = 5.9582, P = 0.0026; pairwise comparisons in Table S4).

Figure 3 Richness and abundance of sub-Antarctic freshwater macroinvertebrates.

Mean richness and mean abundance (±SE) of freshwater macroinvertebrates associated to the four studied lagoons from the Róbalo river watershed, Navarino Island, Southern Chile, during 2015. Bold a, b, c and d indicate significant differences in mean richnnes, italic a and b indicate significant differences in mean abundance (p < 0.05).

Altitudinal community structure

The freshwater macroinvertebrate assemblage showed variability along the gradient, although there were common elements through different elevations. In Bandera lagoon, 700 m a.s.l., 78.22% of the assemblage was represented by the freshwater amphipod genus Hyalella, with oligochaetes contributing 14.8% (Table 4, Fig. 4). In a similar fashion, in El Salto lagoon, Hyalella amphipods and oligochaetes were the most frequent taxa (49.67% and 24.68%, respectively; Table 4, Fig. 4). In Róbalo lake, 250 m a.s.l., amphipods represented almost half of the community (46.72%), while oligochaetes, leptophlebiid mayflies, biting and non-biting midges, leeches, and lymnaeid snails comprised the remaining representative fauna (Table 4, Fig. 4). The abundant Hyalella amphipods (30.59%), non-biting midges, lechees, diving beetles and water boatmen were the most abundant taxa in Castor lagoon, 20 m a.s.l. (Table 4, Fig. 4). Significant differences in macroinvertebrate community composition were apparent along the elevation gradient, between the habitats and in interactions between factors (PERMANOVA, Elev F = 39.695, P = 0.0002; Hab F = 5.2073, P = 0.0002; Elev × Hab F = 1.4874, P = 0.0002, Table S5). Pairwise comparisons detected differences between all elevations and habitats, with the exception between submerged vegetation and two other habitats (pairwise comparison, rock bottoms P = 0.1376, aquatic mosses P = 0.0852, Table S6).

Table 4 Sub-Antarctic freshwater macroinvertebrate community composition.

Similarity Percentages (SIMPER) analysis of freshwater macroinvertebrate community composition that contribute approximately 90% of the abundance of the assemblage from sampling sites of the Róbalo river watershed, Navarino Island, Chile.

Elevation (m.a.s.l.)	Species	Av. Abund.	Sim/SD	Contrib. %	Cum. %	
20	Hyalella sp.	5.42	5.78	30.59	30.59	
Chiro-morph1	2.66	1.1	11.64	42.22	
Glossi-morph1	2.1	2.52	11.35	53.57	
Chiro-morph2	2.16	1.13	10.05	63.62	
Lancetes angusticollis	1.91	2.6	8.76	72.39	
Sigara sp.	2.06	1.57	7.31	79.69	
Orth-morph1	2.21	1.01	6.68	86.38	
Tany-morph1	1.92	1.2	6.01	92.39	
250	Hyalella sp.	2.73	3.53	46.72	46.72	
Oligo-morph1	0.88	0.9	8.82	55.54	
Meridialaris chiloeense	0.8	0.59	7.62	63.16	
Ceratopog-morph1	0.47	0.55	6.76	69.92	
Glossi-morph1	0.42	0.65	4.99	74.91	
Chiro-morph1	0.66	0.47	4.83	79.75	
Glossi-morph2	0.86	0.41	4.38	84.12	
Chiro-morph2	0.61	0.65	4.32	88.44	
Pectinidens diaphanum	0.52	0.53	3.46	91.9	
	Hyalella sp.	4.3	4.7	49.67	49.67	
480	Oligo-morph1	2.52	2.27	24.68	74.35	
	Lancetes angusticollis	1.51	1.2	13.16	87.51	
	Chiro-morph1	1.06	0.67	5.4	92.92	
700	Hyalella sp.	4.52	3.37	78.22	78.22	
Oligo-morph1	1.33	1.31	14.8	93.01	
Notes.

Abreviations Av. Abund Average Abundance

Sim/SD Similarity Standard Deviation

Contrib. Contribution

Cum. Cumulative

Chiro Chironomidae

Daph Daphnidae

Glos Glosiphoniidae

Oligo Oligochaeta

Orth Orthocladiinae

Tany Tanypodinae

morph morphotype

Figure 4 Sub-Antarctic freshwater macroinvertebrate community composition.

SIMPER contribution percentages of sub-Antarctic freshwater macroinvertebrates associated to lagoons from the Róbalo river watershed on Navarino Island during 2015. Elevation of each lagoon is indicated in pie chart. Abreviations, Chiro, Chironomidae; Daph, Daphnidae; Glos, Glosiphoniidae; Oligo, Oligochaeta; Orth, Orthocladiinae; Tany, Tanypodinae; morph, morphotype.

Functional feeding groups (FFG) comprised six main categories: collector-filterers, collector-gatherers, collector-predators, herbivores (grazers, shredders and/or piercers), predators, and scrapers. Collector-gatherers and predators where present in every lagoon, although with varying representativeness. Collector-gatherers were the dominant group at all elevations, with representativeness increasing with elevation, ranging from 48.7% at 20 m a.s.l. to 99.7% at 700 m a.s.l. (Table 5, Fig. 5). Predators comprised almost a quarter of the FFG assemblage at each elevation, excepting a marked decrease at 700 m a.s.l. (to 0.3%). Herbivores, collector-filterers, collector-predators and scrapers were present at almost every elevation, but rarely represented more than 10% of the total assemblage (Table 5, Fig. 5). Similar to community composition, significant differences were found in FFG composition with elevation, between habitats and in their interaction (PERMANOVA, Elev F = 25.168, P = 0.0002; Hab F = 4.6319, P = 0.0002; Elev x Hab F = 2.1298, P = 0.0002, Table S7). Differences between all elevations and habitats were detected by pairwise comparisons, excepting between submerged vegetation and two other habitats (pairwise comparison, rock bottoms P = 0.0544, aquatic mosses P = 0.3382, Table S8).

Table 5 Sub-Antarctic freshwater macroinvertebrates functional feeding groups.

Similarity Percentages (SIMPER) analysis of freshwater macroinvertebrate functional feeding groups composition that contribute approximately 90% of the abundance of the assemblage from sampling sites of the Róbalo river watershed, Navarino Island, Chile.

Elevation (m.a.s.l.)	FFG	Av. Abund.	Sim/SD	Contrib. %	Cum. %	
20	C-G	5.71	7.55	48.67	48.67	
Pred	2.86	8.00	24.53	73.2	
Herb	2.06	1.65	12.64	85.84	
C-pred	1.93	1.13	9.30	95.14	
250	C-G	2.98	3.5	55.91	55.91	
Pred	1.57	1.56	21.52	77.43	
Herb	1.23	0.96	14.04	91.47	
480	C-G	4.56	5.28	71.94	71.94	
Pred	1.84	1.74	23.49	95.43	
700	C-G	4.61	8.19	99.66	99.66	
Notes.

Abreviations FFG Functional feeding groups

Av. Abund. Average Abundance

Sim/SD Similarity Standard Deviation

Contrib. Contribution

Cum. Cumulative

C-G collector gatherers

C-fil collector filterers

C-pred collector predators

Herb herbivores

Pred predators

Figure 5 Sub-antarctic freshwater macroinvertebrates functional feeding groups.

SIMPER contribution percentages of sub-Antarctic freshwater macroinvertebrate functional feeding groups associated to lagoons from the Róbalo river watershed on Navarino Island during 2015. Elevation of each lagoon is indicated in pie chart.

Discussion

The variation of water temperature across spatio-temporal gradients is one of the key abiotic drivers that shape the distribution, ecology and biology of freshwater macroinvertebrates (Vannote et al., 1980). Latitudinal and altitudinal gradients are recognized as ecological analogues and drivers of environmental gradients, with both being considered proxies of temperature variation (Dos Santos et al., 2018). As expected, water temperature showed a marked decrease associated with increasing elevation, while snow and ice cover had longer duration at higher elevations. Similar to altitudinal studies in streams of the Róbalo river watershed (Contador et al., 2015), the lagoon water temperatures reported here dropped markedly over a short geographical distance. This short and steep gradient, and associated environmental clines, stand out when compared to similar magnitude changes in, for example, the Andean region (Jacobsen, 2004; Scheibler, Claps & Roig-Juñent, 2014), Patagonian streams (Miserendino & Pizzolón, 2000; Miserendino & Pizzolón, 2003), or in the Northern Hemisphere Rocky Mountains (Hauer et al., 1997). Several studies have focused on temperature and elevation gradient effects on freshwater macroinvertebrates in South America (Nieto et al., 2016; Shah et al., 2017; Dos Santos et al., 2018). Nevertheless, austral and remote regions, such as Tierra del Fuego and the Cape Horn Biosphere Reserve, have received little attention in this regard (Contador, Kennedy & Rozzi, 2012).

In terms of abiotic features other than temperature, lagoons in the CHBR have a circum-neutral pH, with the exception of the low elevation Castor lagoon (20 m a.s.l.), whose acidic pH is likely caused by the adjacent peat bog complex. Conductivity shows a similar trend, with low values in the higher altitude lagoons and higher values at low elevations, again likely to be related to the surrounding peat bog and associated organic matter input. Moorman et al. (2006) noted that low conductivity values indicate a lack of pollution in these CHBR freshwater systems, and Mach et al. (2016) reported very low to undetectable levels of inorganic pollutants.

Macroinvertebrate groups recorded in this study with the highest species richness were Diptera and Trichoptera, especially chironomid midges and limnephilid caddisflies, while Hyalella amphipods were the most abundant taxon throughout the gradient. Similar composition patterns have previously been described in Patagonian and Magellanic sub-Antarctic streams (Miserendino & Pizzolón, 2000; Contador et al., 2015), and also in the Argentinean Andes (Scheibler, Claps & Roig-Juñent, 2014). Palma (2013), divides Chile into four macroclimatic zones, the arid north, the Mediterranean center, the temperate south, and Patagonia, which may underlie mayfly, stonefly and caddisfly distribution patterns. Nonetheless, in the Patagonian ecoregion two contrasting climatic zones can be identified, an arid steppe region east of the Andes mountain range and hyperhumid forests and wetlands south-west of this mountain range. The latter is known as the Magellanic sub-Antarctic ecoregion (Rozzi et al., 2006). Besides having an understudied freshwater biodiversity, climatic conditions in this ecoregion support a highly endemic but relatively low diversity macroinvertebrate fauna and, at the same time, the region includes the southern latitudinal limit for several insect orders.

Our results show a low-plateau then monotonic decline in richness with increasing elevation, which indicates that community composition is simpler and, in this case, more even (Pielou’s index) towards higher altitude. This pattern differs from the mid-peak trend found in streams of the same watershed (Contador et al., 2015). Similar tendencies have also been reported in Andean, and central and southern Argentinian streams (Corigliano et al., 1996; Miserendino & Pizzolón, 2000; Miserendino & Pizzolón, 2003; Scheibler, Claps & Roig-Juñent, 2014). Elevation-restricted taxa associated with the studied lagoons may have particular potential in research addressing the biotic responses to global and regional climate change. For example, in our study daphniid cladocerans, small hydroporine dytiscids, corixids and the southernmost distributed dragonfly, Rhionaeschna variegata, were restricted to the low elevation sampled lagoons, and are also not present in streams of the same watershed (Contador et al., 2015). Whilst widespread in the stream elevation gradient (Contador et al., 2015; Contador & Kennedy, 2016), limnephilid caddisflies and leptophlebiid mayflies were restricted to mid elevations, where they were common in lagoon assemblages.

This study confirms that elevation has strong effects on community structure, function and environmental features, even in a relatively low elevation mountain range. The harsh environmental conditions of the Magellanic sub-Antarctic ecoregion, which include long periods of ice and snow cover, as well as chronically low temperatures, reduces the opportunity for development thereby increasing development time, as well as habitat availability and food supply, supporting simple but well adapted communities.

The data obtained provide a descriptive baseline of ecological features of macroinvertebrate communities associated with lentic freshwater ecosystems in sub-Antarctic Chile. In conjunction with previous stream studies, it provides a watershed-scale platform of information that is a necessary underpinning for future long-term research in the region. The Magellanic sub-Antarctic ecoregion stands out as an ideal natural laboratory in which to explore how biota will respond to different global environmental change scenarios.

Conclusions

The short (1,000 m) and accessible altitudinal gradient, associated with strong decreases in air and water temperatures, makes the Magellanic sub-Antarctic region an ideal natural laboratory to understand ecological responses to environmental gradients and changes. This study confirms that freshwater macroinvertebrate diversity and community features in this region are strongly influenced by temperature gradients.

Supplemental Information

Supplemental Information 1 Supplementary tables

Supplementary table 1. Abiotic parameters. Supplementary table 2. Taxonomic macroinvertebrate list.Supplementary table 3. Analyses of diversity metrics. Supplementary table 4. Post-hoc comparisons of diversity metrics. Supplementary table 5. Analyses of macroinvertebrate composition. Supplementary table 6. Post-hoc comparisons of macroinvertebrate composition.Supplementary table 7. Analyses of macroinvertebrate functional feeding groups. Supplementary table 8. Post-hoc comparisons of macroinvertebrate functional feeding groups.

Click here for additional data file.

The authors thank the Sub-Antarctic Biocultural Conservation Program for logistics and support in Navarino Island, Chile. This work is a contribution of the Wankara Sub-Antarctic and Antarctic Freshwater Ecosystems Laboratory, Universidad de Magallanes, Puerto Williams, Chile.

Additional Information and Declarations

Competing Interests

Author Contributions

Field Study Permissions

Data Availability

The authors declare there are no competing interests.

Javier Rendoll Cárcamo conceived and designed the experiments, analyzed the data, contributed reagents/materials/analysis tools, prepared figures and/or tables, authored or reviewed drafts of the paper, approved the final draft.

Tamara Contador analyzed the data, contributed reagents/materials/analysis tools, authored or reviewed drafts of the paper, approved the final draft.

Melisa Gañán contributed reagents/materials/analysis tools, prepared figures and/or tables, authored or reviewed drafts of the paper, approved the final draft.

Carolina Pérez Troncoso and Alan Maldonado Márquez analyzed the data, prepared figures and/or tables, approved the final draft.

Peter Convey, James Kennedy and Ricardo Rozzi contributed reagents/materials/analysis tools, authored or reviewed drafts of the paper, approved the final draft.

The following information was supplied relating to field study approvals (i.e., approving body and any reference numbers):

Field and sampling permits were provided by Omora Ethnobotanical Park, and field collection of specimens were approved by Universidad de Magallanes (certificate number: no 80/CEC/2018).

The following information was supplied regarding data availability:

Data is available at the Environmental data initiative: Rendoll Cárcamo JA, Contador TA, Gañán M, Pérez Troncoso CA, Maldonado Márquez AA, Convey P, Kennedy JH, Rozzi R. 2019. Freshwater macroinvertebrate community composition and diversity data from lagoons on Navarino Island, Chile. Environmental Data Initiative. https://doi.org/10.6073/pasta/2e25347a2467d80a19bc87857a092b6a.

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
