# Peer review of "Altitudinal gradients in Magellanic sub-Antarctic lagoons: the effect of elevation on freshwater macroinvertebrate diversity and distribution"

_PeerJ, doi:10.7717/peerj.7128_

## Round 0.1 · original submission · Minor Revisions

Thank you for a very well prepared and interesting paper. The three referees make several complimentary remarks and provide some suggestions for you to consider to improve it.

Reviewer 1 ·

Basic reporting

This article is written in a easy and clear form, and have the structure of a cientific investigation. It uses the appropiate literatura, and demostrate that the cientific group dominates the topic. The figures that they used are very good.
Please write always et al in italics.
The figure 2 indicates temperatura between february 2015 to march 2016, it must be congruent with the methogology described (there, the autor indicates from March 2015-abril 2016).
All the proposed target are well met.

Experimental design

The experimental design is clear and concise, and it leads to obtaining and answering the questions listed
In future, it would be appreciate if the studies include also other chemical parameters, such as MODC (material organic disolved colored), ortophosphate, phosphorous total, Nitrogen total, NH4, NO3, NO2. We know that with all antropic activities, the values ot this parameters will increase in time, so it is necessary to have a baseline.

Validity of the findings

The statistics test are well used and are appropiate to answer the questions.

Additional comments

I congratulate the authors because of the effort shown in this study, and they have shown that good results can be achieved with basic tools.

·

Basic reporting

This is an interesting and very nice study in which authors assessed freshwater macroinvertebrates communities in lentic environments along an altitudinal gradient in southern South America. The manuscript is well written, clear and concise, figures are high quality and methods are technically sound. I strongly recommend it for publication.

Experimental design

It’s a descriptive study. Methods are adequate.

Validity of the findings

Authors provide a description of richness and composition of macroinvertebrates communities along the altitudinal gradient examined. They found that elevation has substantial effects on community structure and function. This information is very welcome for the study area.

Additional comments

Some minor comments:

Figure 1: Although not incorrect, I suggest converting geographic units from UTM to degrees (Decimal Degrees or Degrees, Minutes, Seconds) for easy referencing, since the authors express the location of the Magellanic Sub-Antartic region using degrees (introduction: lines 69 to 72). This would apply also to coordinates provided in Table 1.

Table 2: Please check the superscript in cm2.

Tables 4 & 5: I suggest including the meaning of column headings within the abbreviations in the legends.

·

Basic reporting

The text is unambiguously written in technically correct English. Background, relevant literature and article structure is appropriate. Figures and tables conform to an acceptable format.

The results are only partly relevant to fulfill the aims because of reduced sampling coverage (please see experimental design).

Experimental design

The research is original and falls within the aims and scope of the journal. Its goal is to describe and compare aquatic macroinvertebrate communities along altitudinal gradients on Magellanic sub-Antarctic lagoons. Four lakes/ponds along an altitudinal gradient of 680 m (lower site 20m, highest 700m) were sampled in one occasion. The relevance of the goal relates to the harsh and poorly known environment sampled, and constitutes a great effort to fill the gap in basic knowledge of the area. Nevertheless the number of gradients (transects) and sites (and probably sampling dates too, depending on hydrologic regime) should be increased to attain the objectives. Random noise caused by absence of replication and reduced sample size (number of sites, dates and gradients) could cause spurious or inconsistent results that do not conform to the regional diversity pattern.

Validity of the findings

Methods are described with sufficient detail except the following points:
1) Biological sampling consisted of a "standard sampling" (1 min of net sweeping in four predefined habitats) and a "complementary collection" to "get a comparable and representative sample of each habitat". All samples were analyzed together. The standard sampling is comparable between sites, since they were standardized by time (minutes of net sweeping) but are not readily comparable with other studies that mainly use densities (standardized by area). The complementary collection may vary between sites so even inside the reduced sampling universe of this study, they are not comparable. The authors should compare sites only if similar sampling effort were carried out.
2) The method followed to assign each taxon to its corresponding functional feeding group (FFG) is not mentioned. For example, in the results, figure and table it can be read that the mayfly found in this study (Meridialaris chiloeensis) was assigned to the shredder category, but published works (e.g., Diaz Villanueva & Modenutti 2004, DOI: 10.1002/iroh.200310694) have determined Meridialaris chiloeensis to be a grazer. The assignation of remaining taxa should be also checked. Two ways of facing the assignation of taxa are commonly used: 1) assignation by bibliography (e.g, Rodriguez et al 2011 Rev. Biol. Trop 59: 1537-1552 ), and 2) gut content analyses (e.g., Tomanova et al. 2006, Hydrobiologia, 556, 251-264). If the first is the case it is recommendable to use regional literature since geographic variations in diets are commonly reported. The second approximation is undoubtedly more accurate, since any particularity of the regional fauna or differences between instars of a given taxon can be readily acknowledged. The manuscript should be completed with this information. An introductory paragraph about the relevance of FFG to attain the proposed objectives would be welcomed as additional background.
Conclusions would gain strength and wider validity if more gradients (and sites) are added to the study.

---

## Round 0.2 · accepted · Accept

Thank you for attending to the minor corrections noted by the referees and addressing their comments.